# Detection, Identification and Molecular Characterization of the 16SrII-V Subgroup Phytoplasma Strain Associated with *Pisum sativum* and *Parthenium hysterophorus* L.

**DOI:** 10.3390/plants12040891

**Published:** 2023-02-16

**Authors:** Yi-Ching Chiu, Pei-Qing Liao, Helen Mae Mejia, Ya-Chien Lee, Yuh-Kun Chen, Jun-Yi Yang

**Affiliations:** 1Institute of Biochemistry, National Chung Hsing University, Taichung 402, Taiwan; 2PhD Program in Microbial Genomics, National Chung Hsing University and Academia Sinica, Taichung 402, Taiwan; 3Department of Plant Pathology, National Chung Hsing University, Taichung 402, Taiwan; 4Institute of Biotechnology, National Chung Hsing University, Taichung 402, Taiwan; 5Advanced Plant Biotechnology Center, National Chung Hsing University, Taichung 402, Taiwan

**Keywords:** phytoplasma, *Pisum sativum*, *Parthenium hysterophorus* L., witches’ broom, phyllody, virescence

## Abstract

Two unrelated plant species, green pea and parthenium weed, harboring typical phytoplasma symptoms, were discovered in Yunlin, Taiwan. Green pea (*Pisum sativum.*) and parthenium weed (*Parthenium hysterophorus* L.) are both herbaceous annual plants belonging to the Fabaceae and Asteraceae families, respectively. Displayed symptoms were witches’ broom, phyllody and virescence, which are typical indications of phytoplasma infection. Pleomorphic phytoplasma-like bodies were observed under the transmission electron microscope in the sieve elements of symptomatic green pea and parthenium weed. The *i*PhyClassifier-based virtual RFLP study demonstrated that the phytoplasma associated with the diseased plants belongs to the 16SrII-V subgroup. The disease symptoms of both plants can be explained by the identification of PHYL1 and SAP11 effectors, identical to those of peanut witches’ broom phytoplasma. The phytoplasma strains identified in this study present a very close phylogenetic relationship with other 16SrII-V subgroup phytoplasma strains discovered in Taiwan. These results not only convey the local status of the 16SrII-V subgroup phytoplasma strains but also encourage attention to be given to preventing the spread of this threat before it becomes pervasive.

## 1. Introduction

Phytoplasmas are phytopathogenic bacteria restricted to phloem sieve elements [1,2]. They are known to infect several hundred species and spread by phloem-feeding insect vectors, such as leafhoppers, planthoppers and psyllids [3]. Symptoms caused by phytoplasma infection include general stunting, purple top (anthocyanin accumulation), dwarfing, wilting, vivipary (premature germination), yellowing, witches’ broom (proliferation of shoots with small leaves), virescence (abnormal green pigmentation), phyllody (development of leaf-like structures instead of regular flowers) and general declines, which cause a major loss in agricultural production and quality [4,5].

Since its identification in 1967 by Japanese scientists, studies on phytoplasma have been limited to the fact that it cannot be cultured [6]. Thus, appropriate treatments have not been available due to the lack of an in-depth understanding of its biology. However, diagnoses and elucidation have advanced in the last decades with the help of biotechnological techniques on the molecular level, and this is already a remarkable leap [2,5]. The plethora of superficial symptoms of phytoplasma infection is mainly caused by secreted effectors that disturb the developmental processes of host plants [7]. So far, four secreted effectors are known to cause the abnormalities observed in diseased plants: SAP54/PHYL1, SAP11, SAP05 and TENGU [8,9,10,11,12,13,14].

*Pisum sativum* or green pea is a herbaceous annual plant in the family Fabaceae. It is native to the Mediterranean Basin and the Near East and is grown almost worldwide as edible seeds. This species has a climbing hollow stem with terminal tendrils reaching up to 2–3 m in length. It is starchy and packed with fiber, protein, vitamin A, vitamin B6, vitamin C, vitamin K, phosphorus, magnesium, copper, iron, zinc and lutein, and is high in phytochemical substances which exhibit medicinal properties, e.g., phenolics, terpenoids and nitrogenous compounds [15]. In Taiwan, the major diseases affecting pea production are bacterial, fungal and viral infections, e.g., pea fusarium wilt caused by *Fusarium oxysporum* f. sp. pisi, pea root rot caused by *Fusarium solani* f. sp. Pisi and mosaic disease of pea caused by lettuce mosaic virus [16,17].

*Parthenium hysterophorus* (L.), commonly known as parthenium weed, is an annual herbaceous weed of the family Asteraceae that originated in Northeast Mexico and eventually spread to Africa, Australia and Asia [18]. It is a highly invasive weed that causes massive agricultural loss and health hazards for humans and livestock, such as contact dermatitis, respiratory allergies and, in more extreme cases, cell mutagenicity [19,20]. However, contrary to its evinced health threat, *P. hysterophorus* is medicinally utilized by some tribes for various ailments, including inflammation, skin problems, colds and heart troubles [20]. Although it does not serve any economic value, it can be used as an alternative plant host to accelerate the spread of disease, especially since parthenium weed can grow under various environmental conditions [21].

Phytoplasmas belonging to the 16SrII, 16SrVIII, 16SrX, 16SrXI and 16SrXII groups are known to be associated with several crop diseases in Taiwan, e.g., peanut witches’ broom, loofah witches’ broom, pear decline, rice yellow dwarf and papaya yellows [22,23,24,25]. Pea plants infected by phytoplasmas in a variety of groups and subgroups (16SrI, 16SrII-C, 16SrII-D and 16SrXII-A) have been identified in Asia, America and Europe [26]. Parthenium weed is recognized as a natural host of phytoplasma diseases associated with 16SrII-A, 16SrII-C and 16SrII-D subgroups in Asia and Africa [27,28,29]. In this study, the 16SrII-V subgroup phytoplasma strain associated with green peas and parthenium weed in Taiwan are reported for the first time. In addition, SAP11 and PHYL1 effectors are identified to explain the disease symptoms observed in the phytoplasma-infected green pea and parthenium weed.

## 2. Results and Discussion

### 2.1. Green Pea and Parthenium Weed Exhibited Typical Phytoplasma-Infected Symptoms

In the February of 2022, green peas with disease symptoms consistent with phytoplasma infection were found in Mailiao, Yunlin County, Taiwan (Figure 1A). Outward symptoms observed were the abnormal proliferation of branches as shown in Figure 1B, virescence and the development of leaf-like structures in place of regular flowers as shown in Figure 1C,D. Closer anatomical comparison of the floral organs of symptomatic and healthy-looking plants showed that the calyx and petals of symptomatic plants were leaf-like, and the proliferated shoots replaced the carpels (Figure 1E). The pods of symptomatic plants are shriveled and their seeds are abnormally smaller than the healthy-looking one of the cultivated pea variety (Figure 1F). Infected green peas are not consumable and this is considered an agricultural loss. The incidence rate in the area was about 45% based on the observation of a total of 29 plants (13 were symptomatic and 16 were healthy-looking).

In the August of 2021, parthenium weeds displaying outward phytoplasma disease symptoms were also discovered in Mailiao, Yunlin County, Taiwan (Figure 2A). The incidence rate in the sampling area was about 11% (4 were symptomatic and 31 were healthy-looking). Infected samples lose general flower characteristics, and the branches have dramatically proliferated (Figure 2B,C). Closer observation of the flower samples of infected ones revealed definite virescence and phyllody symptoms.

### 2.2. Identification of the Phytoplasma Associated with the Symptomatic Green Pea and Parthenium Weed

Symptomatic leaves of both the green pea and parthenium weed were first analyzed under an electron transmission microscope (TEM) to confirm the presence of phytoplasma in the phloem sieve elements. Indeed, an accumulation of pleomorphic (circular, elliptical and bell-shaped) organisms of about 200 to 800 nm in size was found upon examination (Figure 3A,B). The size and the irregularity of shapes are the typical morphology of phytoplasma. 

Leaves of both the symptomatic green pea and parthenium weed were brought to molecular analysis. Genomic DNAs collected randomly from healthy-looking and symptomatic samples were subjected to a nested polymerase chain reaction (PCR). The 1.2 kb DNA fragments of the conserved 16S ribosomal RNA were amplified from both the symptomatic green pea and parthenium weed samples (Figure 4A). There is no clear amplicon visible in the healthy-looking sample H1 (Figure 4A).

Furthermore, western blotting analyses were conducted using a polyclonal antibody originally raised against the immunodominant membrane protein (Imp) of peanut witches’ broom (PnWB) phytoplasma [30]. As a result, a specific signal with a size of 19 kDa was detected from both the symptomatic green pea and parthenium weed samples, but not from healthy-looking plants (Figure 4B). These results verify that phytoplasma is indeed associated with the witches’ broom diseases found in symptomatic green peas and parthenium weeds.

### 2.3. Classification of the Phytoplasma Associated with the Symptomatic Green Pea and Parthenium Weed

The DNA fragments of the conserved 16S ribosomal RNA amplified from the symptomatic green pea and parthenium weed were purified and subsequently sequenced for classification. The DNA information was deposited in Genbank as partial sequences of the 16S rRNA gene of *Pisum sativum* witches’ broom (PsWB) phytoplasma under accession No. OM827254 and *Parthenium hysterophorus* witches’ broom phytoplasma (PhWB) under accession number OM215201. BLAST analysis confirmed that the 16S rRNA sequences of PsWB phytoplasma and PhWB phytoplasma are identical to that of GenBank accession no. NZ_AMWZ01000008 (complement [31109 to 32640]) of peanut witches’ broom (PnWB) phytoplasma, a 16SrII-V subgroup phytoplasma strain identified in Taiwan [22]. A phylogenetic tree based on the 16S rRNA gene sequence comparison was then generated to illustrate the evolutionary relationship between PsWB and PhWB phytoplasmas with the existing phytoplasmas identified in Taiwan (Figure 5A). As illustrated, PsWB and PhWB phytoplasmas displayed a high sequence identity to other strains within the 16SrII group phytoplasma identified in Taiwan, including strains found in peanut, snake gourd, soybean, mungbean, pear, sweet potato, purple coneflower, threeflower tickclover, *Eclipta prostrata*, *Ixeris chinensis*, *Emilia sonchifolia*, *Digera muricata* and *Nicotiana plumbaginifolia*. Using *i*PhyClassifier, the virtual RFLP pattern of the 16S rRNA sequences of PsWB and PhWB phytoplasmas could be classified into the 16SrII-V subgroup (Appendix A). 

Imp, found on the outermost part of phytoplasma, is unique among phytoplasma strains [31,32]. The DNA fragments of the complete *Imp* gene amplified from symptomatic green pea and parthenium weed were also purified and subsequently sequenced. Then, a phylogenetic tree was generated based on the full-length amino acid sequences of Imp homologs (Figure 5B). Results revealed that the Imp of PsWB and PhWB is identical to that of PnWB and EpWB, the 16SrII-V subgroup phytoplasma strains.

### 2.4. Identification of Putative Effectors Responsible for the Phyllody and Virescence Symptoms Associated with PsWB and PhWB Diseases

Phytoplasma effectors have been identified as causing dramatic changes in the morphology of host plants. As has already been reported, the PHYL1/SAP54 effector degrades the MADS BOX of plants, resulting in phyllody and virescence. DNA fragments encoding PHYL1 were amplified by PCR from both the symptomatic green pea and parthenium weed (Figure 6A). There is no clear amplicon visible in the healthy-looking sample H1 (Figure 6A). EpWB phytoplasma-infected and healthy *C. roseus* were again used as a positive and negative control, respectively. PHYL1 proteins identified from PsWB and PhWB phytoplasmas were identical, and further BLAST analysis revealed that they shared 100% sequence identity with the PHYL1 of PnWB phytoplasma (accession No. WP_004994552) (Figure 7A). PHYL1 homologs were composed of four groups (phyl-A, -B, -C and -D), where members of the phyl-A, -C and -D groups can interact with floral homologous MADS domain transcription factors and decreased the amount of SEP1–4 and AP1 in plants [33]. The PHYL1 identified from PsWB and PhWB phytoplasmas can be classified into the phyl-D group. These results support the phyllody and virescence symptoms observed in symptomatic green peas and parthenium weeds.

SAP11 causes the proliferation of axillary meristems by degrading the class II TCP (TEOSINTE BRANCHED1, CYCLOIDEA, PROLIFERATING CELL FACTOR 1 and 2) transcription factors, particularly the CYC/TB1 (CYCLOIDEA/TEOSINTE BRANCHED 1) clades [8,34]. DNA fragments encoding SAP11 were amplified from both symptomatic green peas and parthenium weed by PCR (Figure 6B). SAP11 protein of PsWB and PhWB phytoplasmas was identical and further BLAST analysis also revealed its identity with the SAP11 of PnWB phytoplasma (accession no. EMR14684) (Figure 7B). As has already been reported, SAP11 identified from the PnWB phytoplasma has the ability to disrupt TB/CYC-TCP transcription factors [8]. These results support the witches’ broom symptom observed in symptomatic green peas and parthenium weed. 

## 3. Materials and Methods

### 3.1. Field Sampling

Both symptomatic parthenium weeds and green peas were found in field surveys where peanut fields were adjacent or nearby. In August 2021, symptomatic parthenium weeds were collected in an uncultivated field (23°46′29.8″ N, 120°15′02.3″ E) without management. In February 2022, symptomatic green peas were collected in a cultivated field (23°44′26.3″ N, 120°16′26.8″ E) for family production. Leaf samples from symptomatic plants exhibiting typical phytoplasma symptoms were collected for further examination. Healthy-looking (asymptomatic) plants from the same fields were also taken for a negative control. For genomic DNA (gDNA) extraction, samples were stored at −80 °C until use.

### 3.2. Polymerase Chain Reaction (PCR) and Nested PCR

The Plant Genomic DNA Purification Kit (DP022-150, GeneMark) was used to extract the gDNA from the leaves of healthy-looking and symptomatic plants. For the green pea, 1 healthy-looking and 7 symptomatic plants were analyzed; for the parthenium weed, 2 healthy-looking and 4 symptomatic plants were analyzed. The 16S ribosomal RNA (rRNA) gene was examined by nested PCR using the phytoplasma universal primer pairs P1/P7 followed by R16F2n/R16R2 [35,36,37,38]. The first round of nested PCR was carried out for 12 cycles in a final volume of 20 μL, in which 1 μL of the product was used as a template for the second round of PCR executed for 35 cycles. For DNA sequencing, the P1/P7 primer pair-amplified DNA fragments were sequenced with P1 and a nested primer 5′-GGGTCTTTACTGACGCTGAGG-3′ [39]. For PCR analyses, specific primer sets for *Imp*, *PHYL1* and *SAP11* genes according to the genome information of the ‘*Ca*. P. aurantifolia’ NCHU2014 (accession No. CP040925) and ‘*Ca*. P. aurantifolia’ NTU2011 (accession No. NZ_AMWZ01000001.1-13.1) were designed. Healthy and phytoplasma (16SrII-V)-infected *C. roseus* were used as a negative and positive control, respectively. The primer sequences are listed in Appendix A.

### 3.3. Transmission Electron Microscopy (TEM) Assay

The TEM procedure was conducted as described previously with modification [40]. Symptomatic samples were fixed with 2.5% glutaraldehyde in 0.1 M phosphate buffer (pH 7.2) and then with 1% osmium tetraoxide. Subsequently, samples were dehydrated with an ethanol series and immersed in LR White Resin. Samples were cut using an ultramicrotome and stained with uranyl acetate and lead citrate. Then, ultrathin sections were observed under a JEOL JEM-1400 series 120 kV TEM (Jeol, Tokyo, Japan). The Gatan Orius SC 1000B bottom mounted CCD-camera (Gatan Inc., Pleasanton, CA, USA) was used for photo collection. 

### 3.4. Western Blotting

Samples were ground in liquid nitrogen and then directly added with 2.5x SDS sample buffer (5 mM EDTA, 5% SDS, 0.3 M Tris-HCl, pH 6.8, 20% glycerol, 1% β-mercaptoethanol and bromophenyl blue). The total cell extracts were prepared by heating at 95 °C in a dry bath for 10 min. After centrifugation at 13,000× *g* for 12 min, supernatants were obtained and proteins were separated using SDS-PAGE. Polyclonal antibodies against Imp and PHYL1 were used to monitor protein amounts [30,41]. Western blotting was performed using enhanced chemiluminescence western-blotting reagents (Amersham), and chemiluminescence signals were captured using ImageQuant LAS 4000 Mini (GE Healthcare).

### 3.5. Phylogenetic Tree Construction

MEGA-X software was used for phylogenetic tree construction based on the sequence comparisons of 16S rRNA gene, or Imp homologues from different phytoplasma species. Multiple sequence alignments were performed using the ClustalW program. Then, molecular evolutionary analyses were performed by the neighbor-joining method with bootstrapping. The numbers at the branch points are bootstrap values representing the percentages of replicate trees based on 1000 repeats.

### 3.6. iPhyClassifier Analysis

Virtual RFLP patterns were generated by in silico digestion of the 1.2 kb DNA fragment (R16F2n/R16R2) of the 16S rRNA gene identified from *Pisum sativum* witches’ broom phytoplasma (accession No. OM827254) and *Parthenium hysteroporus* witches’ broom phytoplasma (accession No. OM215201) using *i*PhyClassifier, an interactive online tool (https://plantpathology.ba.ars.usda.gov/cgi-bin/resource/iphyclassifier.cgi, accessed on 1 April 2022) [42]. 

## 4. Conclusions

In this study, phytoplasma associated with green peas and parthenium weed is the first record in Taiwan for both plants. The intensive molecular analysis proved that green peas and parthenium weed were infected with the 16SrII-V subgroup phytoplasma strains, which exhibit a very close phylogenetic relationship with the phytoplasma strain associated with peanuts. Because symptomatic parthenium weeds and green peas were found in field surveys with peanut fields adjacent or nearby, it is imperative to give attention to how this disease is treated or prevented. The epidemiological cycle of phytoplasma involves plants and insects. More than parthenium weed, several invasive weeds, e.g., *Eclipta prostrata*, *Ixeris chinensis*, *Digera muricata* and *Nicotiana plumbaginifolia,* were also found to be infected with the 16SrII-V subgroup phytoplasma strains in Mailiao, Yunlin, Taiwan [43,44,45,46]. Thus, identification of insect vectors is required to make informed decisions in order to manage the phytoplasma diseases, for they pose a potential threat to Taiwan’s agricultural industry and food security.

## Figures and Tables

**Figure 1 plants-12-00891-f001:**
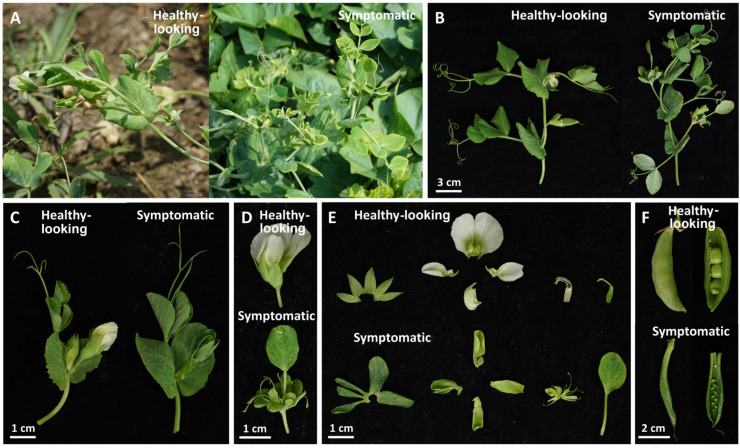
Comparison of the phenotype of symptomatic and healthy-looking green peas. (**A**) The healthy-looking and symptomatic plants found in the field of green peas in Mailiao, Yunlin, Taiwan. (**B**) The shoot proliferation with varying severity from floral organ symptoms observed in symptomatic peas. (**C**) A flower with phyllody and virescence symptoms in the symptomatic green pea (right) compared to the healthy-looking one (left). (**D**) Close-up view of flared flowers in the healthy-looking green pea (above), and green leaf-like flowers in the symptomatic green pea (below). (**E**) An anatomical diagram of healthy-looking (above) and symptomatic (below) floral organs showing sepal, petal, stamen and carpel from left to right. (**F**) A significant reduction in the seed development of the symptomatic green pea compared to the healthy-looking one of the cultivated pea variety.

**Figure 2 plants-12-00891-f002:**
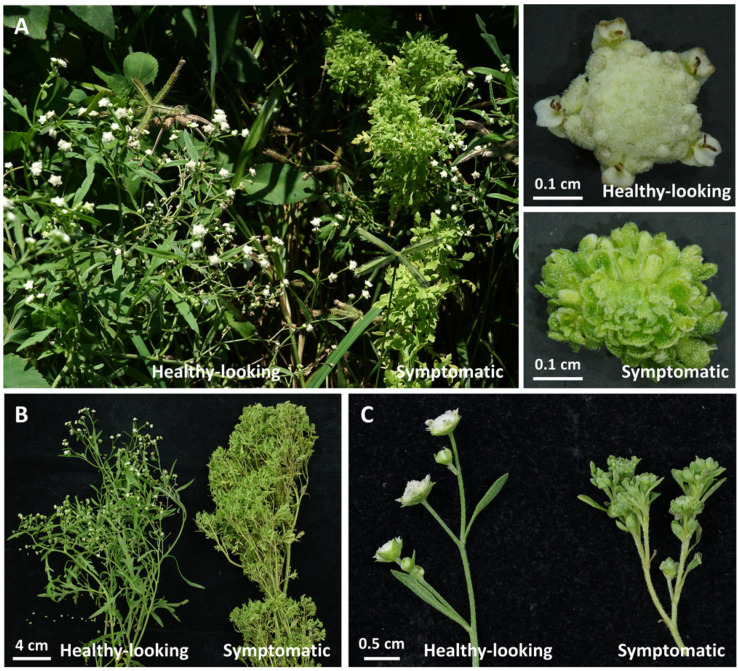
Disease symptoms of phytoplasma-infected parthenium weed. (**A**) Samples of healthy-looking and symptomatic plants collected in Mailiao, Yunlin, Taiwan. (**B**) Photos of a healthy-looking plant with standard branch and a symptomatic plant with dramatic branch proliferation. (**C**) Close-up view of regular flowers with yellow color and symptomatic flowers with phyllody and virescence symptoms. Enlarged images of a healthy-looking and symptomatic flower are present in the upper-right corner.

**Figure 3 plants-12-00891-f003:**
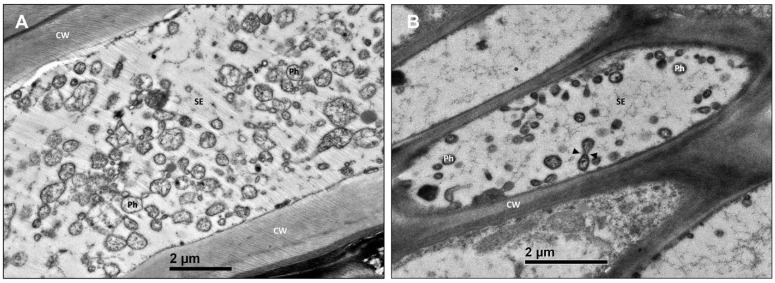
Transmission electron microscope examination of symptomatic samples. Micrographs of phytoplasma bodies within leaf veins of the symptomatic green pea (**A**) and parthenium weed (**B**). Black arrowheads indicate phytoplasma undergoing cell division. The sieve elements were occupied by pleomorphic phytoplasma bodies. CW, cell wall; Ph, phytoplasma; SE, sieve element.

**Figure 4 plants-12-00891-f004:**
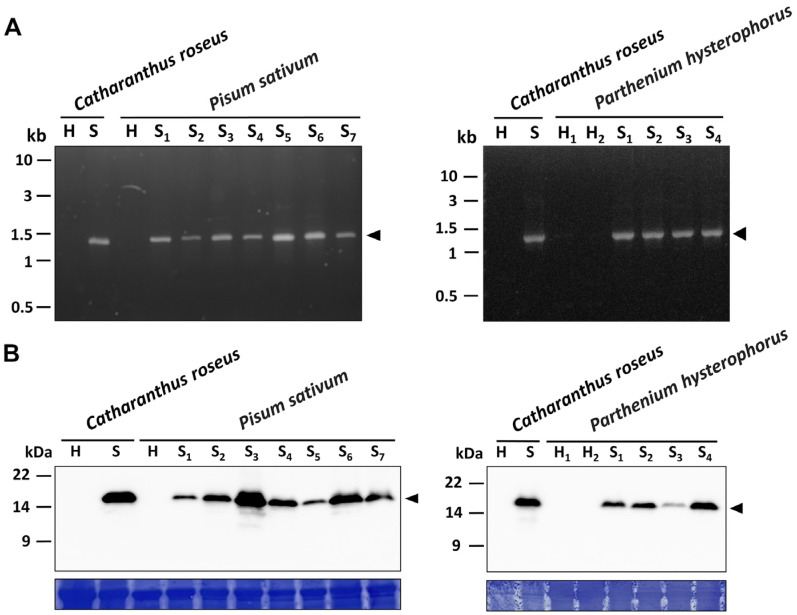
PCR and western blotting analyses of symptomatic *Pisum sativum* (green pea) and *Parthenium hysterophorus* (parthenium weed). (**A**) The 1.2 kb DNA fragment (arrowhead) of the 16S rRNA gene was amplified by PCR from symptomatic (S) samples and not from healthy-looking (H) ones. *Catharanthus roseus* infected by the 16SrII-V subgroup phytoplasma strain (*Echinacea purpurea* witches’ broom, EpWB) was used as a positive control. (**B**) Western blotting analyses using polyclonal antibody against immunodominant membrane protein (Imp). The expected signal of 19 kDa (arrowhead) specific for Imp was detected in symptomatic samples and not in healthy-looking (H) ones (upper panel). The large subunit of Rubisco was visualized using Coomassie Brilliant Blue staining (lower panel) and was used as a loading control.

**Figure 5 plants-12-00891-f005:**
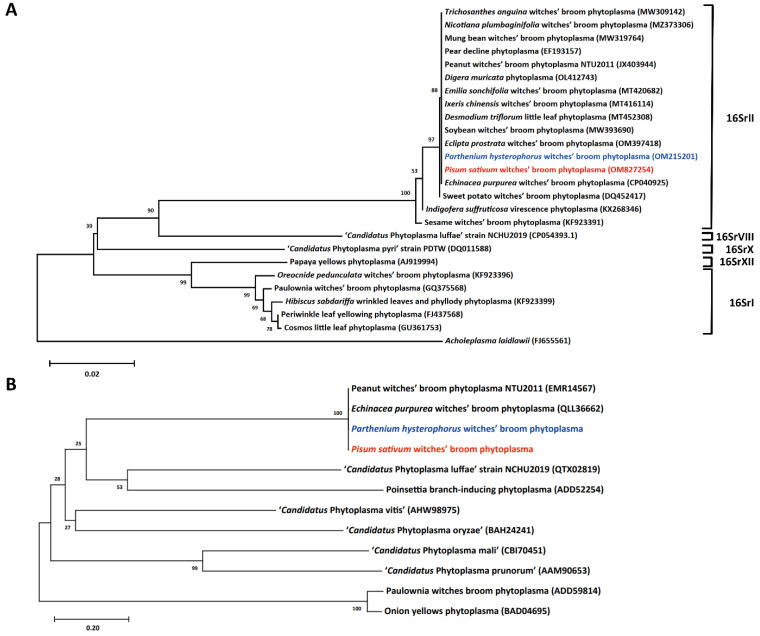
Phylogenies of representative phytoplasmas based on the 16S rRNA genes and Imp proteins. (**A**) The phylogenetic tree was generated based on the R16F2n/R16R2 fragment of the 16S rRNA gene from phytoplasma strains identified in Taiwan. *Acholeplasma laidlawii* served as an outgroup. (**B**) The phylogenetic tree was generated using phytoplasma Imp homologs. In both panels, the 16S rRNA genes and Imp proteins of *Pisum sativum* witches’ broom phytoplasma and *Parthenium hysterophorus* witches’ broom phytoplasma identified in this study are presented in red and blue, respectively. Numbers next to internal branches indicate the bootstrap support levels based on 1000 re-sampling.

**Figure 6 plants-12-00891-f006:**
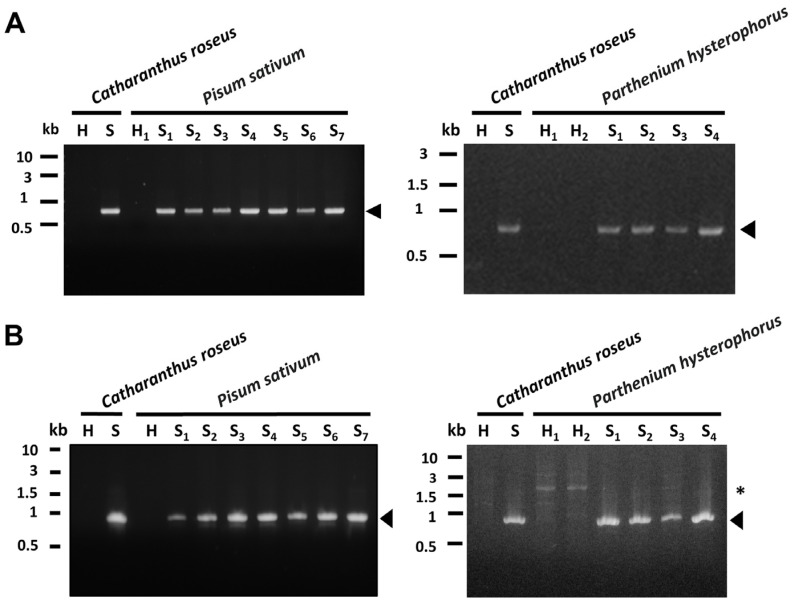
PCR analyses of *PHYL1* and *SAP11* genes from phytoplasma-infected *Pisum sativum* (green pea) and *Parthenium hysterophorus* (parthenium weed). DNA samples prepared from healthy-looking (H) and symptomatic (S) plants were used for PCR analysis. The 0.7 kb DNA fragment of phytoplasma *PHYL1* gene (**A**) and the 0.9 kb DNA fragment of phytoplasma *SAP11* gene (**B**) are indicated by arrowhead. Non-specific bandings are indicated by asterisk. *Catharanthus roseus* infected by the 16SrII-V subgroup ‘*Ca.* P. aurantifolia’ NCHU2014 was used as a positive control.

**Figure 7 plants-12-00891-f007:**
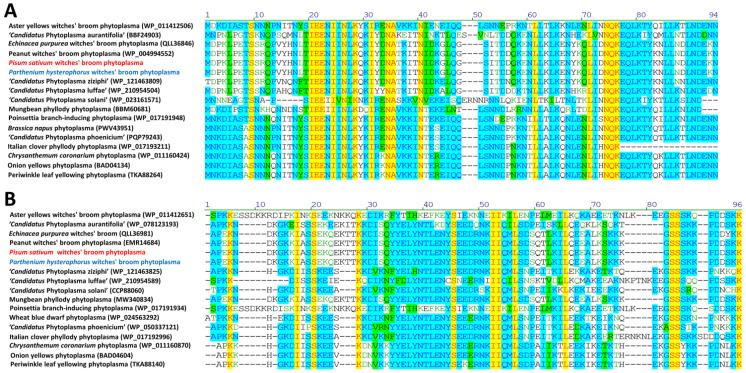
Sequence comparison of effectors responsible for the phyllody and virescence symptoms associated with phytoplasma diseases. Multiple sequence alignment of PHYL1/SAP54 (**A**) and SAP11 (**B**) homologues were generated using AlignX from Vector NTI without signal peptide. Identical residues are shaded in yellow; conservative residues are shaded in blue; blocks of similar residues are shaded in green; weakly similar residues are in green font; and non-similar residues are in black font.

## Data Availability

The data presented in this study are available in this article or the Appendix A.

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
