# Peer review of "Detection, Identification and Molecular Characterization of the 16SrII-V Subgroup Phytoplasma Strain Associated with Pisum sativum and Parthenium hysterophorus L."

_plants, 2023, doi:10.3390/plants12040891_

Round 1

Reviewer 1 Report

The manuscript plants-2120079 deals with the first record in Taiwan of 16SrII-V subgroup phytoplasmas in green pea and parthenium weed. Phytoplasma infection of these species is relevant as they are species of high food interest (green pea) or invasive weed (Parthenium). Consequently, a correct association of the symptoms observed (well described in the work), will allow an adequate monitoring of infections and an accurate assessment of the epidemiological risk.

Thanks to the combination of electron microscopy and molecular biology techniques, authors demonstrated the presence of phytoplasmas showing a very close phylogenetic relationship with the 16SrII-V subgroup phytoplasma strain ('Candidatus Phytoplasma aurantifolia'). All experimental steps are well exposed and carried out; I think the work is also well presented.

I have a single issue that should be clarified regarding the possible spread of the risk of infection. Authors reported 13 symptomatic and 16 healthy plants as numbers. (45% incidence)

Does that mean that the observation was made on a total of 29 plants? Is it therefore an industrial or family production? This should be clarified and highlighted in the discussion, in relation to the risk of diffusion of this new phytoplasma in commercial productions in Taiwan.

In the results and discussion section, detailed information already reported in the materials and methods section is often repeated. They should be eliminated to avoid repetitions. Moreover, primers are listed in the Supplementary Table S1.

Ex.

ü  Genomic DNAs were collected from healthy and symptomatic samples and were subjected to a nested polymerase chain reaction (PCR) using universal primer pairs P1/P7 followed by R16F2n/R16R2. Healthy and Echinacea purpurea witches’-broom (EpWB) phytoplasma (16SrII-V)-infected Catharanthus roseus were used as a negative and positive control, respectivel

ü  DNA fragments encoding PHYL1 were amplified by PCR using primer pair 5’ CCAAAATATGTTAACTCGTGC- 3’/5’-TGTTCATATTATGAAAACTCC-3’ from both symptomatic green pea and parthenium weed

ü  DNA fragments encoding SAP11 were amplified from both symptomatic
green pea and parthenium weed by PCR using primer pair 5’-CGAGACGAAAGACAC-
CAAGAAG-3’/5’-TTGAAACCAACCAACTTATAG-3’

Author Response

We greatly appreciate the comments and suggestions. All points are well taken and addressed accordingly. Please see the attachment.

Reviewer 2 Report

This study provides the characterization of a phytoplasma strain belonging to the 16SrII-V subgroup associated with Pisum sativum and Parthenium hysterophorus in Taiwan. The authors characterized two phytoplasma secreted effectors (SAP11 and PHYL1) to better explain the observed symptoms.

I suggest a few edits or clarifications in order to make some important concepts clear for a broader audience of researchers working with phytoplasma diseases.

I also suggest a few important changes highlighted throughout the text, among those the most important regards the classification of the phytoplasma. The authors reported in Introduction and in other sections of the text that the strain is identified as “16SrII-V subgroup ‘Ca. P. aurantifolia’”. However, I am a little bit concern with this classification. I was not able connect with iPhyclassifier at this moment, but I run a blast with the reference strain for Ca. P. aurantifolia (U15442.1) and the identity is 98.23%. I suggest the author to:

1.     Report results from the classification from iPhyClassifier (Candidatus assignment)

2.     Please revise the last two papers 10.1099/ijsem.0.005353 and https://doi.org/10.3390/biology11081119.

Taking into account all these information, I think the authors should classify it as 16SrII-V subgroup or ‘Ca. P. aurantifolia’ related strain. Also no information about insect vector is available and this is another reason why the assignment to a Candidatus Phytoplasma species should be reported carefully.

Others required edits:

Introduction: there are few suggestions in the area comments in the pdf that I strongly recommend integrating for the sake of clarity.

Material & Methods: I suggest including a paragraph with the description of the habitat of collections, a better description of the symptom recordings and of the plant material analyzed at laboratory.

Results: I appreciated the section 2.1 which provide a detailed report of the symptoms on both cultivated plant and weed. I suggest including a paragraph in material and methods to describe the methods when collecting data to populate this part.

There is an important edit to make regarding the causal agent in paragraph 2.2 (please see explanation in the pdf attached).

I also suggest to re build a phylogenetic tree for 16Sr gene to include more strains from Taiwan in 16SrII to show the diversity. I do not understand why the authors included several strains from Taiwan from the same group which is not the focus of the paper (i.e. group I), if there a reason then it should be better explained in the text, otherwise I suggest to drop out some 16SrI and include more 16SrII.

Discussion: I found online another paper from the authors (a research note: Disease Note First Report of ‘Candidatus Phytoplasma aurantifolia’ Associated with the Invasive Weed Eclipta prostrata (L.) in Taiwan) that report another plant (form the same location?) infected by 16SrII-V which is Eclipta prostrata (also in NCBI OM397418). The strain deposited in NCBI having high percentage of identity with the strains deposited in this manuscript. I suggest the author to cite and compare their previous findings and discuss further the meaning of this broad spreading in the field.

Conclusion: I think that the authors could make a better point about the epidemiological cycle for this disease in Taiwan also mentioning the insect vectors.

Please see all my comments in the attached pdf below.

Author Response

(The authors gave the same response as above.)

Reviewer 3 Report

The manuscript is well written, and the results obtained are interesting for the reader.

Prior to acceptance for publication, authors must provide information on the following points:

- On page 2, the authors talk about 13 symptomatic green pea plants! It is unclear how many of these plants have been infected. The authors characterized only 7 samples found to be infected with 16SrII phytoplasma! Please explain.

-Taxonomically speaking, phytoplasma subgroups are indicated by capital alphabet letters and not Roman numbers. Please check this demarcation.  

Author Response

(The authors gave the same response as above.)

Round 2

Reviewer 2 Report

Dear authors
thank you to improve the version of your manuscript.
